# The competing effects of racial discrimination and racial identity on the predicted number of days incarcerated in the US: A national profile of Black, Latino/Latina, and American Indian/ Alaska Native populations

**George Pro**[1]*, **Ricky Camplain**[2], **Charles H. Lea III**[3]

**1** Southern Public Health and Criminal Justice Research Center, Fay W. Boozman College of Public Health, University of Arkansas for Medical Sciences, Little Rock, Arkansas, United States of America, **2** Department of Health Sciences, Center for Health Equity Research, Northern Arizona University, Flagstaff, Arizona, United States of America, **3** Graduate College of Social Work, University of Houston, Houston, Texas, United States of America

* GCPro@uams.edu

**Data Availability Statement:** The National Epidemiologic Survey of Alcohol and Related

## Abstract

### Objective

Racial discrimination and racial identity may compete to influence incarceration risk. We estimated the predicted days incarcerated in a national US sample of Black, Latino/Latina, and American Indian/Alaska Native (AI/AN) individuals.

### Methods

We used the 2012–2013 National Epidemiologic Survey on Alcohol and Related Conditions-III (n = 14,728) to identify individual incarceration history. We used zero-inflated Poisson regression to predict the number of days incarcerated across racial discrimination and racial identity scores.

### Results

Racial discrimination and identity varied between races/ethnicities, such that racial discrimination exposure was highest among Hispanic individuals, while racial identity was highest among Black individuals. Racial discrimination was positively associated with days incarcerated among Black individuals ($\beta$ = 0.070, p<0.0001) and AI/AN individuals ($\beta$ = 0.174, p<0.000). Racial identity was negatively associated with days incarcerated among Black individuals ($\beta$ = -0.147, p<0.0001). The predicted number of days incarcerated was highest among Black individuals (130 days) with high discrimination scores.

### Conclusion

Racial discrimination and racial identity were associated with days incarcerated, and the association varied by racial/ethnic sub-group. Informed by these findings, we suggest that intervention strategies targeting incarceration prevention should be tailored to the unique

Conditions is available through the National Institute on Alcohol Abuse and Alcoholism. Use is restricted to only those who have been granted access to the dataset through NIAAA. Instructions on how to obtain the dataset are below: https://www.niaaa.nih.gov/procedures-obtaining-dataset.

**Funding:** The authors received no specific funding for this work.

**Competing interests:** The authors have declared that no competing interests exist.

experiences of racial/ethnic minoritized individuals at the greatest risk. Policies aimed at reversing mass incarceration should consider how carceral systems fit within the wider contexts of historical racism, discrimination, and structural determinants of health.

## Introduction

Racial/ethnic minoritized people have borne the burden of the mass incarceration phenomenon, with Black populations reaching imprisonment rates nearly 10 times higher than White populations in several states [1]. While the U.S. incarcerated population has slowly declined since 2011 [2], Black, Latino/Latina, and American Indians/Alaska Native (AIAN) populations remain overrepresented in prisons and jails [3].

Several factors are associated with incarceration risk, including poor health [4], substance use and mental health diagnoses [5], poverty [6], and having an incarcerated parent [7]. One factor in particular–racial discrimination–may also be a risk factor for incarceration. The effect of discrimination on involvement in the criminal justice system is often framed in terms of systemic biases operating on higher-order ecological levels [8], but evidence linking interpersonal discrimination with criminal justice outcomes is scant. Importantly, personal experiences with racial discrimination are associated with several social and health sequelae that are closely related to incarceration [9]. Some research has addressed the links between race/ethnicity and discrimination based on attitudes and perceptions toward felony offenders [10,11], but these findings are limited to post-incarceration experiences of other non-racial types of discrimination.

Black populations in general, and Black men in particular, report high levels of lifetime discrimination [12,13]. Black, Latino/Latina, and AIAN populations report more exposure to discriminatory behavior in health care [14] and educational settings [15,16] than White populations. Furthermore, race-related stress and discrimination have broad deleterious effects on multiple biopsychosocial levels [17].

Research addressing racial identity is notably less common than that of discrimination. Broadly, having a strong racial identity–the significance and meaning an individual places on their race–has been shown to mitigate several negative health effects associated with racial discrimination [18]. For example, strong racial identity has been found to buffer the negative effect of discrimination on serious criminal offending [19], but this report was limited to a sample of Black individuals and did not compare offending between racial/ethnic minority groups.

Racial identity may have a protective effect on incarceration risk. Furthermore, racial identity may diminish part of the deleterious effect of racial discrimination on incarceration outcomes. However, little evidence exists regarding the moderating effects of racial identity across levels of racial discrimination in the context of incarceration. Given this knowledge gap, we sought to better understand differential associations between racial discrimination and incarceration, whether these associations are conditional on levels of racial identity, as well as the extent to which these conditions vary between Black, Latino/Latina, and AIAN groups. Specifically, in this paper we 1) describe a nationally representative epidemiological profile of racial discrimination, racial identity, and incarceration history, and 2) estimate the predicted number of days incarcerated across levels of racial discrimination within low, mid, and high levels of racial identity. We also explore differences in the moderating effect of racial identity by race/ethnicity.

## Materials and methods

### Data source and sample

We used data from the National Epidemiologic Survey of Alcohol and Related Conditions–III (NESARC-III) (April 2012 –June 2013) to assess group differences in incarceration, discrimination, and identity. Our final analytic sample included Black, Latino/Latina, and American Indian/Alaska Native survey respondents (N = 14,728). Given the framing of our study around continued disparities and oppression among these three groups, we did not include White (n = 19,194) or Asian/Native Hawaiian/other Pacific Islander (n = 1,801) respondents in our sample. All racial/ethnic categories were available as a single pre-constructed variable in the original dataset. The development and sampling methods of NESARC-III have been described extensively elsewhere [20]. In short, NESARC-III provides individual-level survey data on topics ranging from substance use and mental health disorders, health services utilization, and many unique social and cultural characteristics. NESARC-III uses a complex sampling design and provides sample weights and strata for use in analyses. Broadly, the survey weights and strata are applied in order to account for the oversampling of housing units in pre-defined high-minority geographic areas. NESARC-III was administered to households within a total of 7,200 segments, and segments were clustered within 150 primary sampling units. Access to NESARC-III was granted to the study team by the National Institute for Alcohol Abuse and Alcoholism. We restricted our analytic sample to Black, Latino/Latina, and AIAN individuals who had complete data for all study variables (n = 14,728).

### Variables

Our dependent variable of interest was the total number of days spent incarcerated throughout the lifetime. We defined total number of days incarcerated as the sum of responses to two questions, including: 1) Total duration (days) in jail or juvenile detention center before age 18, and; 2) Total duration (days) in jail or correctional facility since age 18.

We considered racial/ethnic discrimination and racial/ethnic identity as our focal independent variables. NESARC-III discrimination questions were derived from the Experiences with Discrimination scales developed by Krieger and colleagues [21,22] and have been described elsewhere [23]. Participants were asked six questions about lifetime experiences with racial/ethnic discrimination. Discrimination response options were on a 5-point scale and included: 0 = never; 1 = almost never; 2 = sometimes; 3 = fairly often, and; 4 = very often. Discrimination questions addressed how often the respondent experienced discrimination because of their race/ethnicity: 1) in their ability to obtain health care; 2) in how they were treated when they got care; 3) in public; 4) in any other situation; 5) by being called a racist name, and; 6) by being made fun of, picked on, or threatened. We created a discrimination score that represents the frequency, or magnitude, of exposure to discriminatory behavior. We calculated the sum of the six discrimination questions to achieve a score ranging from 0–24.

The NESARC-III racial/ethnic identity scale has been described elsewhere [23]. Broadly, the NESARC-III scale was adopted from previous tools that broadly assessed an individual's self-concept that derives from their knowledge of, or membership in, a social group [24–26]. Respondents were asked eight questions about their racial/ethnic identity. Identity response options were on a 6-point scale and included: 1 = strongly disagree; 2 = disagree; 3 = somewhat disagree; 4 = somewhat agree; 5 = agree, and; 6 = strongly agree. Identity questions capture the extent to which the respondent: 1) has a strong sense of self as a member of their racial/ethnic group; 2) identifies with other members of their racial/ethnic group; 3) considers most of their close friends to be from their own racial/ethnic group; 4) believes racial/ethnic heritage is

important; 5) is more comfortable in social situations where other members of their racial/ethnic group are present; 6) is proud of their racial/ethnic heritage; 7) believes their racial/ethnic background plays a big part in interaction with others, and; 8) believes their values and behaviors are shared by people of their racial/ethnic background. We created an identity scale score by calculating the sum of all eight identity questions to achieve a score ranging from 0–48.

We also considered several relevant covariates in our analyses, based on *a priori* knowledge of possible confounding in the relationships between incarceration, race/ethnicity, discrimination, and identity. Categorical covariates included sex (male or female); educational attainment (less than high school, high school completion, some college or a two-year degree, or college graduate with a bachelor's degree), and; lifetime drug or alcohol use disorder (yes or no). We condensed several survey response options to create our new 4-level educational attainment variable. Less than high school included those with no formal schooling or those that completed any grade up through 11. High school completion included those who completed grade 12 or received a graduate equivalency degree (GED). Those with some college or a two-year degree were defined as those who had attended a four-year college but did not receive a bachelor's degree, those with an associate's degree, or those with another two-year technical degree. College graduates were defined as those who received a bachelor's degree, attended some graduate or professional studies (completed bachelor's degree but not a graduate degree), or completed a master's degree or equivalent or another higher graduate degree. We also included a rate dependent age variable, defined as the number of days incarcerated divided by the number of years at risk of being incarcerated.

## Analysis

We used survey procedures in SAS (v9.4) where appropriate and incorporated sample weights and strata to account for the parent study's complex sampling design. First, we generated within-group unweighted sample sizes, weighted sample sizes, and weighted percentages for each study variable in the total sample and disaggregated by racial/ethnic group. We also calculated the weighted means for discrimination and identity scores for the total sample and within racial/ethnic groups.

We also estimated the association between discrimination and identity scores and the number of days incarcerated. We grouped discrimination scores (0–24) and identity scores (0–48) into three groups of low, middle, and high based on tertile distributions within each racial/ethnic group. Categorizing into tertiles refers to placing the lowest 33.3% of scores into the low group, the middle 33.3% of scores into the middle group, and the highest 33.3% of scores into the high group. Tertiles were created for identity (Black, low = 0–31, mid = 32–36, high = 37–48; Hispanic, low = 0–32, mid = 33–38, high = 39–48; AI/AN, low = 0–27, mid = 28–33, high = 34–48) and discrimination (Black, low = 0, mid = 0.5–1.5, high = 2–24; Hispanic, low = 0, mid = 0.5–1.5, high = 2–24; AI/AN, low = 0, mid = 0.5–1, high = 1.5–24). Because the majority of our NESARC-III sample were incarcerated for zero days (86%), we used linear regression with days incarcerated defined as following a zero-inflated Poisson (ZIP) distribution [27]. ZIP models solve two regression equations, including 1) a zero logit model that predicts the log odds of being in the zero (no incarceration) group, and 2) a linear model that estimates the predicted days incarcerated. We ran separate models for each race/ethnicity. ZIP models were adjusted for age group, sex, education, and lifetime drug/alcohol use disorder. Finally, to help visualize group differences, we calculated the mean predicted days incarcerated by racial discrimination tertiles and by racial identity tertiles. We plotted mean predicted days at low, mid, and high tertiles for discrimination and identity, and disaggregated our results by race/ethnicity.

## Results

Fourteen percent of the sample had ever been incarcerated (Table 1), and the mean days incarcerated for the full sample was 52. AIAN individuals demonstrated the highest proportion of ever been incarcerated (26%), while Black individuals demonstrated the highest mean days incarcerated (77 days). The mean discrimination score was highest among Latino/Latina individuals (1.9 out of 24) and the mean identity score was highest among Black individuals (32.9 out of 48).

In our fully adjusted ZIP model, racial identity was positively associated with having zero days incarcerated ($\beta = 0.132$, SE = 0.001, p<0.0001), while racial discrimination was negatively associated with having zero days incarcerated ($\beta = -0.199$, SE = 0.001, p<0.0001) (Table 2). Likewise, from the linear portion of the model, racial identity was negatively associated with the total number of days incarcerated ($\beta = -0.110$, SE = 0.000, p<0.0001), while racial discrimination was positively associated with the total number of days incarcerated ($\beta = 0.063$, SE = 0.001, p<0.0001). These trends were mostly consistent between racial/ethnic groups; the association between identity and fewer days incarcerated was strongest among Black

**Table 1. Descriptive characteristics of the National Epidemiologic Survey of Alcohol and Related Conditions-III participants (2012–2013, N = 14,728, weighted N = 635,788,892).**

| Characteristics | Race/ethnicity | | | | | | | | | | | |
|---|---|---|---|---|---|---|---|---|---|---|---|---|
| | **Black** n = 7,445 Weighted n = 26,550,015 | | | **Latino/Latina** n = 6,804 Weighted n = 33,595,359 | | | **American Indian/ Alaska Native** n = 479 Weighted n = 3,433,517 | | | **Total** N = 14,728 Weighted N = 635,788,892 | | |
| | | Weighted | | | Weighted | | | Weighted | | | Weighted | |
| | n | n | col % | n | n | col % | n | n | col % | N | N | col % |
| **Age group** | | | | | | | | | | | | |
| 18–29 | 1,844 | 7,049,603 | 26.5 | 2,011 | 10,155,198 | 30.2 | 91 | 652,852 | 19.0 | 3,946 | 17,857,653 | 28.1 |
| 30–39 | 1,466 | 4,727,924 | 17.8 | 1,723 | 80,044,498 | 23.8 | 95 | 660,456 | 19.2 | 3,284 | 13,392,878 | 21.1 |
| 40–49 | 1,471 | 4,970,152 | 18.7 | 1,360 | 6,591,721 | 19.6 | 91 | 755,639 | 22.0 | 2,922 | 12,317,511 | 19.4 |
| 50+ | 2,664 | 9,802,336 | 36.9 | 1,710 | 8,843,943 | 26.3 | 202 | 1,364,571 | 39.7 | 4,576 | 20,020,850 | 31.4 |
| **% Female** | 4,415 | 14,603,636 | 55.0 | 3,806 | 16,859,041 | 50.2 | 279 | 2,010,899 | 58.6 | 8,500 | 33,473,576 | 52.6 |
| **Education** | | | | | | | | | | | | |
| Less than high school | 1,257 | 4,261,718 | 16.1 | 2,088 | 10,204,026 | 30.4 | 73 | 524,212 | 15.3 | 3,418 | 14,989,956 | 23.6 |
| High school completion or GED | 2,451 | 8,452,645 | 31.8 | 1,921 | 9,439,928 | 28.1 | 136 | 887,946 | 25.9 | 4,508 | 18,780,519 | 29.6 |
| Some college but did not graduate, or received an associate's or technical degree | 2,600 | 9,433,710 | 16.6 | 1,978 | 9,463,447 | 13.4 | 199 | 1,425,340 | 17.4 | 2,025 | 20,322,497 | 14.9 |
| College completion, bachelor's degree or higher | 1,137 | 4,401,941 | 35.5 | 817 | 4,487,959 | 28.2 | 71 | 596,020 | 41.5 | 4,777 | 20,322,497 | 31.9 |
| **Lifetime drug or alcohol use disorder (% yes)** | 1,878 | 6,784,041 | 25.5 | 1,660 | 8,266,116 | 24.6 | 219 | 1,611,499 | 46.9 | 3,757 | 16,661,656 | 26.2 |
| **% Ever incarcerated** | 1,235 | 4,368,515 | 16.5 | 771 | 3,656,413 | 10.8 | 135 | 900,574 | 26.2 | 2,141 | 8,925,502 | 14.0 |
| **Days incarcerated (weighted mean, 95% CLM)** | | 77.5 (64.5, 90.5) | | | 31.4 (23.9, 38.7) | | | 46.4 (26.5, 66.3) | | | 51.5 (44.6, 58.2) | |
| **Racial discrimination score (0–24; weighted mean, 95% CLM)** | | 1.1 (1.0, 1.2) | | | 1.9 (1.8, 2.0) | | | 1.1 (0.8, 1.3) | | | 1.5 (1.4. 1.6) | |
| **Racial identity score (0–48, weighted mean, 95% CLM)** | | 32.9 (32.6, 33.0) | | | 32.8 (32.6, 33.1) | | | 28.9 (28.1, 29.7) | | | 32.6 (32.5, 32.7) | |

**Table 2. Model results for zero-inflated Poisson regression (NESARC-III, 2012–2013).**

| | Black n = 7,445 | | | Latino/Latina n = 6,804 | | | American Indian/ Alaska Native n = 479 | | | Total n = 14,728 | | |
|---|---|---|---|---|---|---|---|---|---|---|---|---|
| *Zero Model* | β | SE | p | β | SE | p | β | SE | p | β | SE | p |
| **Identity** | | | | | | | | | | | | |
| Tertiles 1–3 | 0.086 | 0.001 | <0.0001 | 0.183 | 0.002 | <0.0001 | 0.305 | 0.004 | <0.0001 | 0.132 | 0.001 | <0.0001 |
| **Discrimination** | | | | | | | | | | | | |
| Tertiles 1–3 | -0.202 | 0.001 | <0.0001 | -0.262 | 0.001 | <0.0001 | -0.213 | 0.003 | <0.0001 | -0.199 | 0.001 | <0.0001 |
| *Linear model* | | | | | | | | | | | | |
| **Identity** | | | | | | | | | | | | |
| Tertiles 1–3 | -0.147 | 0.000 | <0.0001 | 0.001 | 0.000 | <0.0001 | 0.093 | 0.000 | <0.0001 | -0.110 | 0.000 | <0.0001 |
| **Discrimination** | | | | | | | | | | | | |
| Tertiles 1–3 | 0.070 | 0.000 | <0.0001 | -0.031 | 0.000 | <0.0001 | 0.174 | 0.000 | <0.0001 | 0.063 | 0.000 | <0.0001 |

respondents ($\beta$ = -0.147, SE = 0.001, p<0.0001) and the association between discrimination and more days incarcerated was strongest among AI/AN respondents ($\beta$ = 0.174, SE = 0.000, p<0.0001).

We also used output generated by the ZIP models to estimate the predicted days incarcerated across levels of racial identity and discrimination (Fig 1A–1D). Racial discrimination was positively associated with predicted days incarcerated for the total sample and across all three racial/ethnic groups. Changes in days incarcerated were most notable among Black and AI/AN respondents. Black respondents with low discrimination exposure had 42 predicted days incarcerated, whereas Black respondents with high discrimination exposure had 130 predicted days incarcerated, or an increase of 209%. Similarly, AI/AN respondents demonstrated an increase of 106% between low and high levels of discrimination. Racial identity was negatively associated with predicted days incarcerated for the total sample and for all three racial/ethnic groups. The biggest change was observed among Hispanic respondents. Those with low levels of racial identity had 37 predicted days incarcerated, while those with high levels of racial identity had 17 days incarcerated, or a decrease of 54%.

## Supplemental results

Pearson's correlation coefficients for all analytic variables are presented in S1 Table. We identified no multicollinearity between predictor variables using a threshold of r<0.80. We also derived the predicted days incarcerated for each individual discrimination and identity survey item, stratified by race/ethnicity and adjusted for age rate, sex, highest grade completed, and alcohol/drug use (S2 Table). We reported the predicted days incarcerated at the lowest response value and the highest response value for each question, as well as the percentage difference between low and high scores. For example, among Black respondents, the predicted days incarcerated for those with no experience with discrimination in healthcare settings was 33.6, whereas the predicted days incarcerated for those that have experienced discrimination in healthcare 'very often' was 66.2, or a 97.0% increase in days incarcerated between 'no' discrimination and 'very often' discrimination.

## Discussion

In this study, we examined racial/ethnic differences in associations between incarceration and racial discrimination and identity. Our findings highlight the carceral implications of exposure to racial discrimination while controlling for the effect of racial identity. Higher racial identity

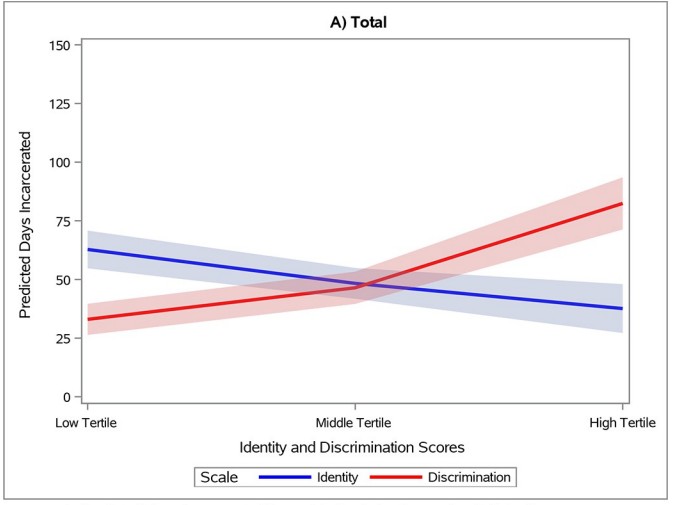

A: Predicted days incarcerated by racial discrimination and racial identity scores
(NESARC-III; Total N=14,728, weighted N=635,788,892)

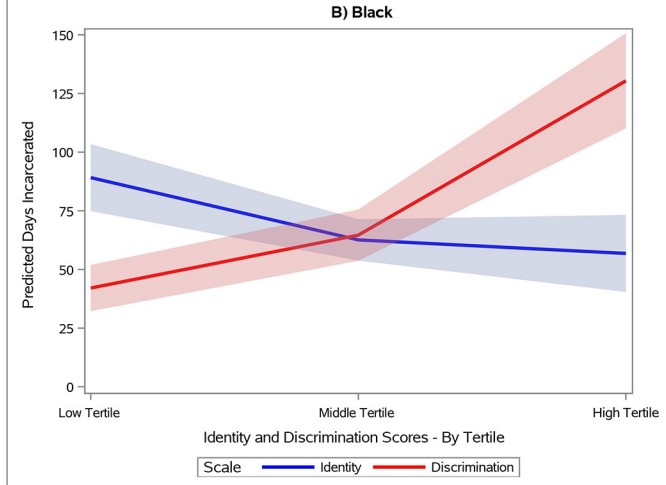

B: Predicted days incarcerated by racial discrimination and racial identity scores
(NESARC-III; Black n=7,445, weighted n=26,550,015)

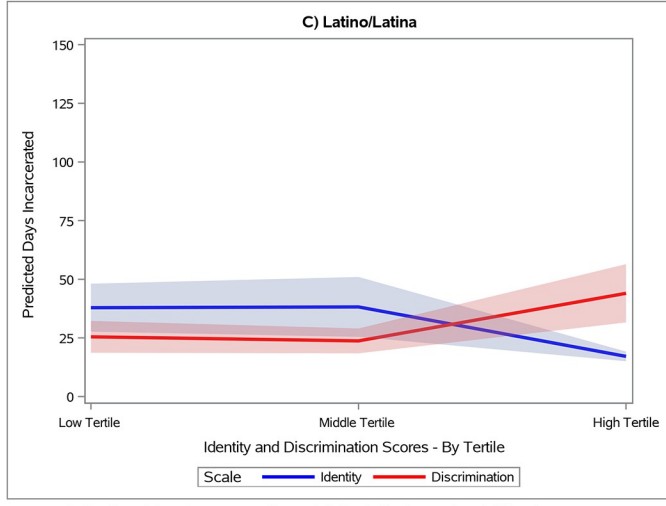

C: Predicted days incarcerated by racial discrimination and racial identity scores
(NESARC-III; Latino/Latina n=6,804, weighted n=33,595,359)

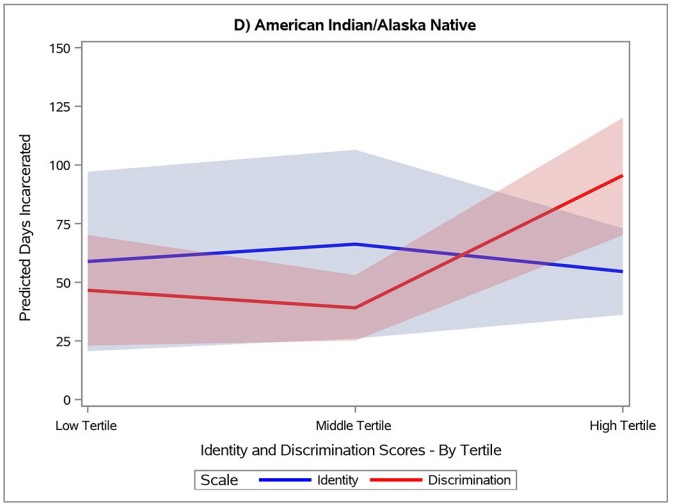

D: Predicted days incarcerated by racial discrimination and racial identity scores
(NESARC-III; American Indian/Alaska Native n=479, weighted n=3,433,517)

**Fig 1. Predicted days incarcerated by racial discrimination and racial identity scores for the total sample and for Black, Latino/Latina, and American Indian/Alaska Native subgroups.** A: Predicted days incarcerated by racial discrimination and racial identity scores (NESARC-III; Total N = 14,728, weighted N = 635,788,892). B: Predicted days incarcerated by racial discrimination and racial identity scores (NESARC-III; Black n = 7,445, weighted n = 26,550,015). C: Predicted days incarcerated by racial discrimination and racial identity scores (NESARC-III; Latino/Latina n = 6,804, weighted n = 33,595,359). D: Predicted days incarcerated by racial discrimination and racial identity scores (NESARC-III; American Indian/Alaska Native n = 479, weighted n = 3,433,517).

scores were associated with fewer days incarcerated, which illustrates a partial buffering effect of identity on discrimination. Days incarcerated were highest among AI/AN and Black individuals with high levels of racial discrimination, and the buffering effect of identity appeared to be the strongest among AI/AN individuals. This is generally aligned with previous research that found that racial discrimination had a weaker effect on depression among Black individuals with strong racial identities [18]. Future research may consider identifying the mechanisms underlying the buffering influence of identity on discrimination.

Our findings are in agreement with previous work showing associations between racism, discrimination, and incarceration [28,29]. Importantly, we have added much needed nuance in our measurement of discrimination and identity, and how these metrics relate to a tangible

outcome–the number of days incarcerated. Visualizing these relationships provides the often missing contextual pieces of racism and culture within narratives of criminal justice and public health. While the mean number of days incarcerated was highest among Black respondents (78 days), we also identified that the AI/AN population reported the highest rate (26%) of ever being detained in their lifetime. The impacts of this cannot be understated, especially given the impact that prior incarceration has on job application and employment opportunities [30], discrimination and stigma based on conviction status [31], and adverse experiences among children of incarcerated parents [32]. By directly addressing social determinants of health across multiple racial/ethnic minoritized groups, the current study illustrates a measurable outcome of structural racism in an easily understood and relatable metric of days spent in a jail or prison setting.

Strategies aimed at reducing incarceration should consider the nuances of discrimination and identity, as well as how they differ between racial/ethnic groups. In particular, interventions taking place within lower ecological levels may serve to weaken the effects of discrimination by empowering those who experience it. For example, empowerment-based approaches that support inclusive communities and teach resiliency and coping may help to advance health equity while diminishing the clout of discriminatory individuals and institutions. In addition, targeting higher ecological levels may move the national discourse towards changing social norms and supporting policies, practices, and built environments that engender racial equity.

## Limitations

NESARC-III considers lifetime incarceration as ever having been in jail or prison. However, jails and prisons differ in many ways, and racial discrimination and identity may affect the likelihood of incarceration differently between the two locations. Exposure to racial discrimination may also differ within jails versus prisons. Future research may highlight more granular effects by differentiating between jails and prisons.

Respondents self-reported their experiences with ever being detained in a juvenile detention center or jail in their lifetime. People may be reluctant to disclose all or part of their criminal histories to interviewers, resulting in the possibility of introducing desirability bias to the incarceration estimates. Such a bias would likely result in a more conservative underestimation of the reported days incarcerated, and it is unknown whether there is variability in underreporting between racial/ethnic groups. To our knowledge, there exist no other national jail datasets that have sample sizes and complex survey designs analogous to NESARC-III. The Bureau of Justice Statistics (BJS) estimated that 2–3% of the US population had been detained in a jail in 2019 [33]. In comparison, our estimate that 14% of adults reported ever being detained is reasonable, given that lifetime estimates require recalling many more years than were included in the BJS 2019 report [33]. Furthermore, regional and state-level jail datasets mostly report on aggregate annual admissions–national data on lifetime estimates of days detained at the individual level does not exist elsewhere. Our finding that 14% of adults had ever been detained in their lives is reasonable. Thus, NESARC-III is an imperfect but useful tool to estimate trends in incarceration.

Geographic data, including state, is not available in NESARC-III. We were therefore unable to test external, state-level variables that likely affect incarceration risk. Our inability to include metrics of structural racism like residential segregation, racial/ethnic population densities, policies around health and social services, or criminal justice and policing was a limitation of this study.

Finally, the cross-sectional nature of our data makes inferences of causality impossible. Rather, we are limited to conclusions of associations of non-temporal events. Future

longitudinal studies would be well positioned to detect clear and causal relationships between structural racism, discrimination, and identity and subsequent incarceration. Importantly, experiences of incarceration may also shape later perceptions of discrimination and identity, further justifying our support for longitudinal study designs.

## Conclusion

Racial discrimination and identity varied between races/ethnicities, such that both discrimination and identity were highest among Black individuals. Black individuals also demonstrated the highest mean days incarcerated across the lifespan, which was more than double the next highest group of AIAN individuals. We also found a strong relationship between discrimination score and days incarcerated. Most notably, racial/ethnic minority groups who had large proportions of their members with the highest discrimination score–Black and AIAN–were estimated to be incarcerated for a total of 9–10 weeks throughout their lives.

Policies aimed at reversing the trend of mass incarceration should consider how our carceral systems fit within the wider contexts of historical racism and structural determinants of health. We recommend addressing the challenges of discrimination on multiple ecological levels. Examples include educational programs emphasizing the values of racial and cultural differences, community-level campaigns organizing for equitable access to health and financial services, and electoral support for representatives who campaign on evidence-based criminal justice reform.

## Supporting information

**S1 Table. Pearson's correlation coefficient matrix.**
(DOCX)

**S2 Table. Predicted days incarcerated by individual discrimination and identity survey items.**
(DOCX)

## Author Contributions

**Conceptualization:** George Pro, Ricky Camplain, Charles H. Lea III.

**Data curation:** George Pro.

**Formal analysis:** George Pro, Ricky Camplain.

**Methodology:** George Pro, Ricky Camplain, Charles H. Lea III.

**Resources:** George Pro, Charles H. Lea III.

**Visualization:** George Pro, Charles H. Lea III.

**Writing – original draft:** George Pro.

**Writing – review & editing:** George Pro, Ricky Camplain, Charles H. Lea III.

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
