## [Decision Letter · Decision Letter 0]

16 Oct 2021

PONE-D-21-10832The competing effects of racial discrimination and racial identity on the predicted number of days incarcerated in the USPLOS ONE

Dear Dr. Pro,

Thank you for submitting your manuscript to PLOS ONE. After careful consideration, we feel that it has merit but does not fully meet PLOS ONE’s publication criteria as it currently stands. Therefore, we invite you to submit a revised version of the manuscript that addresses the points raised during the review process. Please address all comments included in the two referee reports.

We look forward to receiving your revised manuscript.

Kind regards,

Yann Benetreau, PhD

Senior Editor

PLOS ONE

Journal Requirements:

2. Please include a caption for figure 1.

Reviewers' comments:

Reviewer's Responses to Questions

**Comments to the Author**

1. Is the manuscript technically sound, and do the data support the conclusions?

Reviewer #1: Yes

Reviewer #2: Partly

2. Has the statistical analysis been performed appropriately and rigorously? 

Reviewer #1: Yes

Reviewer #2: No

3. Have the authors made all data underlying the findings in their manuscript fully available?

Reviewer #1: Yes

Reviewer #2: Yes

4. Is the manuscript presented in an intelligible fashion and written in standard English?

Reviewer #1: Yes

Reviewer #2: Yes

5. Review Comments to the Author

Reviewer #1: The authors conducted a study of racial discrimination, racial identity, and incarceration risk in a national US sample of Black, Latino/Latina, and American Indian/Alaska Native individuals. Overall, the study is well written, interesting, and will be an important contribution to the literature. The methods are appropriate, thorough, and well-described. I only recommend some minor edits to improve clarity and language.

1. Overall, I recommend not using the term "Blacks", and instead using "Black populations", "Black individuals", or "Black participants".

2. When you discuss the survey procedures and design effects, I'd recommend a sentence on what the clusters and strata represent (like groups of counties, cities, etc.).

3. For the last sentence that "These trends were consistent between racial/ethnic groups, and were strongest among AIAN", I would revise to drop the "and were strongest among AIAN", as it looks like Black and Latinx participants had the strongest association between discrimination and incarceration.

4. Some discussion on how racial identity buffers against internalized racism is recommended.

5. The distinction between jails and prisons noted in the limitations is important; I recommend an additional sentence on how racial discrimination may differ between them.

Reviewer #2: Review of The competing effects of racial discrimination and racial identity on the predicted

number of days incarcerated in the US

General comments

As I read the definition of “discrimination,” the authors’ description of it seems imprecise. As they operationalize it, it is not a measure of discrimination per se, but a measure of “number of types of discrimination ever encountered and recorded in the self-report.” I am not sure that I agree with the decision to disregard the data in the survey on the frequency with which discrimination was encountered. In revision, I think the authors should have to both: a. justify this decision and b. report on analyses in which they created a variable(s) that took account of the frequency data. This comment is based on my belief that people who experience discrimination “all the time” are likely to react in different ways than those who experience it “almost never.” The authors may also want to explore whether one particular type of discrimination is a better predictor. After all, with an N of 14 thousand, such analyses should be possible.

Similar explorations should be made and reported with different ways to define racial identity. Number of types of identification ignores the issue of strength of identity. And it is also of interest to know if one particular kind of identification is more or less associated with incarceration.

Analytically, if the dependent variable is number of days of incarceration, then instead of treating age as a confounder, it should be used to create a rate dependent variable that is defined as “number of days incarcerated divided by numbers of years at risk of being incarcerated.” This is because I expect that self-reports of discrimination; racial identity; and number of days incarcerated are all correlated with age. This is certainly true for days of incarceration. The other variables are likely related because of one or both of the following: a. generations have different experiences and beliefs (see Mannheim’s analysis of generations) and/or b. life stage factors. This seems to me to be a very serious error.

I am skeptical of the unvalidated claim that 86% of the sample reported zero days of incarceration. First, these are self-reported data, and people may be reluctant to disclose criminal histories. Second, even if they are willing to disclose, they might minimize it; or they might say “never incarcerated” in order to avoid follow up questions (depending on skip patterns) and shorten the interview. To be credible, this figure should be supported by comparisons with other studies (such as other surveys) that do not focus on alcoholism but perhaps on criminal victimization and other surveys like some NORC has done.

In an analysis such as this, reporting on the correlations among the main analytic data (probably for total, and within racial categories) is needed. Particularly concerning is the possibility that racial identity and racial discrimination experiences could conceivably be very high.

I think that the Conclusions should include a strong statement about the much higher rates of “ever incarcerated” among AIAN. Given the many limitations that accrue to simply reporting that you were ever arrested in job applications and other critical processes, this is of enormous impact.

Discussion and Conclusions discuss structural racism a lot. This makes the analytic failure to include any measures of structural racism stand out as a major defect and limitation. (See below under “important additional comments.”

Important additional comments

1. Perhaps the title should indicate the restricted nature of the sample that was analyzed by adding “among Black, Latino/Latina, and AIAN individuals.” Although the exclusion of whites is not problematic given the theoretical framing and the realities of oppression, the exclusion of Asian Americans is an important limitation and should be noted in the title (I think) and should be justified in Methods.

2. Limitations should include the inability to analyze how locality or workplace influenced incarceration. Measures of structural racism like racial/ethnic population densities, racial/ethnic residential and occupational segregation are almost certainly associated with probability of getting arrested, and likely with both racial discrimination and racial identity.

Minor Comments

1. line 2 “bared” is misleading. Probably “borne” would be best

2. line 5 – 6 “remain…remains” I will stop text editing at this point. It is clear that serious text editing is needed.

3. l 9 I am not sure “disenfranchisement” is the best term for this.

4. p. 5 l 6 – 7: Here or in Methods, the variable that was used to limit the sample to Black, Latino/Latina, and AIAN individuals should be described.

5. Table 1 and first paragraph of Results: Important to state whether mean days incarcerated is for the total analytic sample or only for the subset who reported non-zero on this variable.

6. p. 13 -14: I am not sure that the claim about structural racism in the following sentence is justified by the data (although I think it fully justified in reality): “This study supports the notion that incarceration is driven, in part, by the mechanisms of structural racism and personal discriminatory acts towards racial/ethnic minorities.” I say this because I see no measures of structural racism in the set of independent variables.

6. PLOS authors have the option to publish the peer review history of their article (what does this mean?). If published, this will include your full peer review and any attached files.

Reviewer #1: No

Reviewer #2: No

---

## [Author Response · Author response to Decision Letter 0]

9 Feb 2022

We thank the journal editors and our two reviewers for the opportunity to revise our manuscript. We received invaluable insight from our reviewers about study design, more accurately defining discrimination and identity, and further exploring the influence of discrimination on incarceration risk. We believe our manuscript has been substantially improved as a result of the thoughtful consideration put into each review and is now suitable for publication in PLOS One. We have responded to each comment in detail below, and have submitted a revised manuscript with changes tracked in MS Word. 

Reviewer 1

General comments

1. The authors conducted a study of racial discrimination, racial identity, and incarceration risk in a national US sample of Black, Latino/Latina, and American Indian/Alaska Native individuals. Overall, the study is well written, interesting, and will be an important contribution to the literature. The methods are appropriate, thorough, and well-described. I only recommend some minor edits to improve clarity and language.

Response: Thank you for your critical feedback and positive support for our research. We have responded to each of your comments below.

2. Overall, I recommend not using the term "Blacks", and instead using "Black populations", "Black individuals", or "Black participants".

Response: We appreciate this feedback as we are constantly striving to be conscious of the language we use to describe disparities. We have gone through the manuscript and edited how we describe groups, in line with using “Black individuals” or “American Indian/Alaska Native respondents”. 

Methods

3. When you discuss the survey procedures and design effects, I'd recommend a sentence on what the clusters and strata represent (like groups of counties, cities, etc.).

Response: This is a good opportunity to provide more information about the survey design to our readers. The probability sampling procedure of NESARC-III is complex, including primary sampling units, secondary sampling units, segments, dwelling units, and respondents. We added two sentences in our methods section outlining the purpose and process of oversampling in high-minority areas, and direct the readers to the NESARC technical documentation for further information. 

Results

4. For the last sentence that "These trends were consistent between racial/ethnic groups, and were strongest among AIAN", I would revise to drop the "and were strongest among AIAN", as it looks like Black and Latinx participants had the strongest association between discrimination and incarceration.

Response: Please see our response to Comments #7 and #8. Our definitions for discrimination and identity have changed to include the full information available on the frequency of discrimination exposure and the magnitude of agreement with identity statements. As such, our regression estimates have changed moderately and our results section has been updated, including the sentence described in the comment above.

Discussion

5. Some discussion on how racial identity buffers against internalized racism is recommended.

Response: Better understanding the mechanisms underlying the buffering characteristics of racial identity is an important future research question that came out of this current study. For this study, we believe that speculating on how identity may be operating to suppress the effect of discrimination exposure is beyond the scope of what we measured. However, we did add several sentences to our discussion section outlining how our findings align with previous research addressing the intersection of identity and discrimination. 

Limitations

6. The distinction between jails and prisons noted in the limitations is important; I recommend an additional sentence on how racial discrimination may differ between them.

Response: We have added a sentence about the importance of considering how racial discrimination may vary between the two locales.

Reviewer 2

General comments

7. As I read the definition of “discrimination,” the authors’ description of it seems imprecise. As they operationalize it, it is not a measure of discrimination per se, but a measure of “number of types of discrimination ever encountered and recorded in the self-report.” I am not sure that I agree with the decision to disregard the data in the survey on the frequency with which discrimination was encountered. In revision, I think the authors should have to both: a. justify this decision and b. report on analyses in which they created a variable(s) that took account of the frequency data. This comment is based on my belief that people who experience discrimination “all the time” are likely to react in different ways than those who experience it “almost never.” 

Response: Thank you for this very helpful comment. We agree that our original definition of discrimination was imprecise – our single category of ‘any’ discrimination included a wide range of experiences, from almost never to all the time. This variable failed to capture what we are trying to measure, which is the frequency, or magnitude, of exposure to discriminatory behavior. 

We replaced the original ‘any/none’ variable with a new variable that takes into account the frequency of discrimination exposure. The new composite scale is based on six survey questions, each with response options of 0 (never) to 4 (very often), resulting in a single scale of 0 to 24. Scores at the higher end of the spectrum result from multiple responses endorsing often or very often exposure. This new approach allows for more granularity in measuring discrimination, as higher scores reflect a greater frequency of discrimination. 

8. Similar explorations should be made and reported with different ways to define racial identity. Number of types of identification ignores the issue of strength of identity.

Response: Our original racial/ethnic identity scale was similar to the original discrimination scale, in that we defined ‘any’ positive identity as any respondent that somewhat agreed, agreed, or strongly agreed with each statement. The original variable was based on eight survey questions and the scale ranged from 0 to 8. However, applying the same logic as the reviewer raised above in Comment #7, experiences of individuals who ‘somewhat agree’ are likely different than those who ‘strongly agree’. Therefore we reconstructed our identity variable to reflect the magnitude, or strength, of affirming responses. The revised identity scale now ranges from 0 to 48, with higher scores reflecting a greater magnitude of agreement with each question. 

9. The authors may also want to explore whether one particular type of discrimination is a better predictor. After all, with an N of 14 thousand, such analyses should be possible.

And it is also of interest to know if one particular kind of identification is more or less associated with incarceration.

Response: This is a great point, as our composite discrimination and identity scales do not capture the influence of each individual sub-item on incarceration. We created a Supplemental Table 1, which outlines the relationship between each survey item and incarceration. Specifically, we derived the predicted days incarcerated for the lowest and highest scores for each question, and also reported the percentage difference in days. For example, among Black respondents, the predicted days incarcerated for those with no experience with discrimination in healthcare settings was 33.6, whereas the predicted days incarcerated for those that have experienced discrimination in healthcare ‘very often’ was 66.2, or a 97.0% increase in days incarcerated between ‘no’ discrimination and ‘very often’ discrimination. 

Title

10. Perhaps the title should indicate the restricted nature of the sample that was analyzed by adding “among Black, Latino/Latina, and AIAN individuals.” Although the exclusion of whites is not problematic given the theoretical framing and the realities of oppression, the exclusion of Asian Americans is an important limitation and should be noted in the title (I think) and should be justified in Methods.

Response: We have changed our title to state specifically the groups included in this study, and have added text in the methods section about how we arrived at our final analytic sample. Our new title is “The competing effects of racial discrimination and racial identity on the predicted number of days incarcerated in the US: A national profile of Black, Latino/Latina, and American Indian/Alaska Native populations”. 

Introduction

11. Line 2 “bared” is misleading. Probably “borne” would be best

Response: We have changed ‘bared’ to ‘borne’.

12. Line 5 – 6 “remain…remains” I will stop text editing at this point. It is clear that serious text editing is needed.

Response: We appreciate the opportunity to make our writing as clear as possible for our readers. In this particular case, we have respectfully kept the word ‘remain’, as it refers to the plural noun ‘populations’. We have reviewed our manuscript thoroughly for grammatical errors and typos, and have edited sentence structure to be as clear and correct as possible.

13. Line 9 I am not sure “disenfranchisement” is the best term for this.

Response: We have changed ‘one form of disenfranchisement’ to ‘one factor in particular’. 

Methods

14. Analytically, if the dependent variable is number of days of incarceration, then instead of treating age as a confounder, it should be used to create a rate dependent variable that is defined as “number of days incarcerated divided by numbers of years at risk of being incarcerated.” This is because I expect that self-reports of discrimination; racial identity; and number of days incarcerated are all correlated with age. This is certainly true for days of incarceration. The other variables are likely related because of one or both of the following: a. generations have different experiences and beliefs (see Mannheim’s analysis of generations) and/or b. life stage factors. This seems to me to be a very serious error.

Response: This is a great point and a methodological oversight on our part. While we still report on age groups in our descriptive Table 1, we have replaced age group with a rate dependent variable in our predictive models. 

15. I am skeptical of the unvalidated claim that 86% of the sample reported zero days of incarceration. First, these are self-reported data, and people may be reluctant to disclose criminal histories. Second, even if they are willing to disclose, they might minimize it; or they might say “never incarcerated” in order to avoid follow up questions (depending on skip patterns) and shorten the interview. To be credible, this figure should be supported by comparisons with other studies (such as other surveys) that do not focus on alcoholism but perhaps on criminal victimization and other surveys like some NORC has done.

Response: Recognizing possible bias in responses to sensitive questions about incarceration history is critical for this study, and not including this as a limitation was an oversight. We have included a paragraph in our limitations section outlining how the possibility of desirability bias could affect the reported estimates of incarceration. In addition, we compared our findings to BJS jail admission data, and concluded that while NESARC-III is imperfect, it is a useful and validated tool to estimate population characteristics, including experiences of incarceration. 

16. In an analysis such as this, reporting on the correlations among the main analytic data (probably for total, and within racial categories) is needed. Particularly concerning is the possibility that racial identity and racial discrimination experiences could conceivably be very high.

Response: This was helpful for us to think through the complex relationships between all of our analytic variables. We created a new Supplemental Table 2 which includes all of the information from a Pearson’s correlation matrix, for the total sample and stratified by race/ethnicity. 

17. The variable that was used to limit the sample to Black, Latino/Latina, and AIAN individuals should be described.

Response: We have expanded our description of the race/ethnicity variable we used. NESARC=III provides a single variable that includes pre-constructed categories of race and ethnicity. Please also see our response to Comment #10, where we expanded our description of each racial/ethnic category available in NESARC-III.

Results

18. Table 1 and first paragraph of Results: Important to state whether mean days incarcerated is for the total analytic sample or only for the subset who reported non-zero on this variable.

Response: We appreciate the opportunity to clarify how we are reporting the study results. The mean days incarcerated is for the total sample, and we have clarified this accordingly in the Results section. 

Discussion

19. I think that the Conclusions should include a strong statement about the much higher rates of “ever incarcerated” among AIAN. Given the many limitations that accrue to simply reporting that you were ever arrested in job applications and other critical processes, this is of enormous impact.

Response: Thank you for this comment – we had not discussed this impactful finding in our original submission but we agree that it deserves attention in our revised discussion section. We have added several sentences and three new citations summarizing the impact of such high rates of incarceration among American Indian/Alaska Native respondents. We also brought to our reader’s attention the juxtaposition of Black respondents having the highest mean number of days incarcerated (78 days), while AI/AN respondents had the highest proportion of any lifetime incarceration (26%). 

20. Discussion and Conclusions discuss structural racism a lot. This makes the analytic failure to include any measures of structural racism stand out as a major defect and limitation. 

Response: We value the opportunity to clarify for our readers how a discussion of structural racism fits in to the overall narrative of race, discrimination, and incarceration. We could not identify any variables within the NESARC dataset that closely resembled anything like structural racism. NESARC does not include a variable for state, which would allow researchers to identify other external state-level data that would serve as a proxy indicator of system-level discrimination and racism, like residential segregation, health provider shortage areas, or state-wide financial estimates of disproportionate loan/mortgage rejections among poor and racial minority groups, for example. At the same time, we believe that incarceration itself is a form of structural racism. We therefore feel that it is our responsibility to engage in a conversation about structural racism, and that this conversation is within the scope of our findings about racial discrimination and incarceration. While not directly measured or tested in this study, structural racism is the language that we choose to use when contextualizing the mass incarceration epidemic.

21. I am not sure that the claim about structural racism in the following sentence is justified by the data (although I think it fully justified in reality): “This study supports the notion that incarceration is driven, in part, by the mechanisms of structural racism and personal discriminatory acts towards racial/ethnic minorities.” I say this because I see no measures of structural racism in the set of independent variables.

Response: We agree that this sentence in particular is worded in a way that one may assume we directly tested indicators of structural racism. Even in light of our defense of our discussion around structural racism (see Comment #20), we have removed this sentence to make our study design and focal independent variables as clear as possible. 

Limitations

22. Limitations should include the inability to analyze how locality or workplace influenced incarceration. Measures of structural racism like racial/ethnic population densities, racial/ethnic residential and occupational segregation are almost certainly associated with probability of getting arrested, and likely with both racial discrimination and racial identity.

Response: We have included a new paragraph in our Limitations section that brings to attention the lack of geographic data in NESARC-III, including state. At the very least, access to data about which state the survey respondent resides in would allow for the use of external, state-level indicators of social, environmental, occupational, and economic determinants of health.

---

## [Decision Letter · Decision Letter 1]

21 Mar 2022

PONE-D-21-10832R1The competing effects of racial discrimination and racial identity on the predicted number of days incarcerated in the US: A national profile of Black, Latino/Latina, and American Indian/Alaska Native populations

PLOS ONE

Dear Dr. Pro,

Thank you for submitting your manuscript to PLOS ONE. After careful consideration, we feel that it has merit but does not fully meet PLOS ONE’s publication criteria as it currently stands. Therefore, we invite you to submit a revised version of the manuscript that addresses the points raised during the review process.

We look forward to receiving your revised manuscript.

Kind regards,

Syed Ghulam Sarwar Shah, M.B.B.S., M.A., M.Sc., Ph.D.

Academic Editor

PLOS ONE

Journal Requirements:

Additional Editor Comments (if provided):

ABSTRACT: 

Page 2, Lines 11-12: change ‘discrimination exposure’ to ‘racial discrimination exposure’.

Page 2, Line 15: What AI/IN stand for? please spell out these acronyms and other acronyms on their first appearance.

Page 2,Line 19: The conclusion should be based on / refer to the results, which show that Racial discrimination and racial identity are associated with incarceration.

INTRODUCTION

Page 3, line: Please revise ‘upwards of 10 times higher’. It should be either ‘upwards’ or ‘higher’.

Page3,  lines 21-22: Please provide a citation to support the statement: Eighty-five percent of Blacks and 51% of Latino/Latinas study participants in California reported ever being treated unfairly because of their race/ethnicity, compared of Whites participants.’

METHODS

Page7, lines-9-10: Please check the statement: ‘educational attainment (less than high school, high school completion, some college, or college graduate), what’s the difference between ‘some college’ and or ‘college graduate’? These could be merged in one category.

Page 7, lines11-12:  Please refer to your statement: “We also included a rate dependent age variable, defined as the number of days incarcerated divided by the number of years at risk of being incarcerated.” What is 'rate dependent age variable? It could be rate of something depending on age. Based on the definition given in the next sentence, it could be named as 'incarceration age rate' or 'incarceration risk rate'.

LIMITATIONS

Page 15, line13: BJS 2019 report. Could you please spell out what BJS stands for? Also provide a citation for the report.

REFERENCES:

Please report journal names in the abbreviated form, where available.

Reviewers' comments:

Reviewer's Responses to Questions

**Comments to the Author**

1. If the authors have adequately addressed your comments raised in a previous round of review and you feel that this manuscript is now acceptable for publication, you may indicate that here to bypass the “Comments to the Author” section, enter your conflict of interest statement in the “Confidential to Editor” section, and submit your "Accept" recommendation.

Reviewer #1: All comments have been addressed

Reviewer #2: All comments have been addressed

2. Is the manuscript technically sound, and do the data support the conclusions?

Reviewer #1: Yes

Reviewer #2: Yes

3. Has the statistical analysis been performed appropriately and rigorously? 

Reviewer #1: Yes

Reviewer #2: Yes

4. Have the authors made all data underlying the findings in their manuscript fully available?

Reviewer #1: Yes

Reviewer #2: Yes

5. Is the manuscript presented in an intelligible fashion and written in standard English?

Reviewer #1: Yes

Reviewer #2: Yes

6. Review Comments to the Author

Reviewer #1: The authors have successfully addressed my concerns. I have no further recommendations regarding the manuscript.

Reviewer #2: This is an excellent revision and an important paper.

I have one suggestion about how to improve the Abstract, though i view it as optional for the authors. As currently written, the Conclusions section of the abstract seems so general as to convey little meaning. I think the last paragraph of conclusions in the text has excellent suggestions for interventions, and these should be the focus of the conclusions in abstract. This will also, in my opinion, lead to more people reading the paper.

7. PLOS authors have the option to publish the peer review history of their article (what does this mean?). If published, this will include your full peer review and any attached files.

Reviewer #1: **Yes: **Rodman Emory Turpin

Reviewer #2: **Yes: **Samuel R Friedman

---

## [Author Response · Author response to Decision Letter 1]

28 Mar 2022

Comments from the editor

ABSTRACT 

Comment 1: Page 2, Lines 11-12: change ‘discrimination exposure’ to ‘racial discrimination exposure’.

Response: We have made this change.

Comment 2: Page 2, Line 15: What AI/IN stand for? please spell out these acronyms and other acronyms on their first appearance.

Response: Thank you for catching this. We have spelled out the acronym for American Indian/Alaska Native at first use, which is in the second sentence of the abstract.

Comment 3: Page 2, Line 19: The conclusion should be based on / refer to the results, which show that Racial discrimination and racial identity are associated with incarceration.

Response: Thank you for helping us better organize our abstract conclusion. We have reframed this paragraph to restate our findings about discrimination and identity. At the same time, we respectfully feel that it is within the scope of this study to reference mass incarceration and racism as drivers of disparities. Later in the paper, our discussion uses these concepts to contextualize increased risk for involvement in the criminal justice system among underrepresented groups. As an epidemiologic study through a public health lens, we believe that failing to articulate the upstream factors affecting incarceration would be a disservice to our audience. In addition, please see Comment 12 below (Reviewer 2). The sentiment of this feedback was that the abstract conclusion should focus more on these same issues we raised in the discussion section, including structural determinants of health and their effect on incarceration and community health. 

INTRODUCTION

Comment 4: Page 3: Please revise ‘upwards of 10 times higher’. It should be either ‘upwards’ or ‘higher’.

Response: We have rephrased this sentence to read, “nearly 10 times higher”

Comment 5: Page3, lines 21-22: Please provide a citation to support the statement: Eighty-five percent of Blacks and 51% of Latino/Latinas study participants in California reported ever being treated unfairly because of their race/ethnicity, compared of Whites participants.’

Response: We have removed this statement.

METHODS

Comment 6: Page7, lines-9-10: Please check the statement: ‘educational attainment (less than high school, high school completion, some college, or college graduate), what’s the difference between ‘some college’ and or ‘college graduate’? These could be merged in one category.

Response: Thank you for the opportunity to clarify our methods and variable creation. We agree that there are multiple ways to combine and rearrange this education variable. In this case, there is an important distinction between ‘some college’ and ‘college graduate’, such that the former refers to some experience with college coursework but did not finish, while the latter refers to those who graduated with a bachelor’s degree. College completion in particular has multiple implications for employment prospects and income, and may be a proxy indicator for overall economic stability. For these reasons, we have respectfully maintained the separation between ‘some college’ and ‘college graduate’. 

Comment 7: Page 7, lines11-12: Please refer to your statement: “We also included a rate dependent age variable, defined as the number of days incarcerated divided by the number of years at risk of being incarcerated.” What is 'rate dependent age variable? It could be rate of something depending on age. Based on the definition given in the next sentence, it could be named as 'incarceration age rate' or 'incarceration risk rate'.

Response: This is a great question and we appreciate the chance to clarify the definition of all of our study variables. We added this variable to our analysis in response to our second round of peer review. One reviewer specifically suggested that we include this variable, named ‘rate dependent age variable’, and we defined it exactly as recommended by the reviewer. Given that both reviewers have provided two separate rounds of feedback, and both are currently satisfied with the methods and variables as they are (below), we are reluctant to make any further changes that may contradict our response to their previous review. 

LIMITATIONS

Comment 8: Page 15, line13: BJS 2019 report. Could you please spell out what BJS stands for? Also provide a citation for the report.

Response: We have included the acronym ‘BJS’ at the first use of the phrase ‘Bureau of Justice Statistics’. We have also made it clear that the citation for the BJS report (included in two sentences) is for citation #33, or Zeng et al. 

REFERENCES:

Comment 9: Please report journal names in the abbreviated form, where available.

Response: We have changed the journal names to the abbreviated form. 

Reviewer #1 

Comment 10: The authors have successfully addressed my concerns. I have no further recommendations regarding the manuscript.

Response: Thank you for your thoughtful and careful consideration of our manuscript.

Reviewer #2

Comment 11: This is an excellent revision and an important paper.

Response: Thank you for your time and energy put into this critical peer review.

Comment 12: I have one suggestion about how to improve the Abstract, though I view it as optional for the authors. As currently written, the Conclusions section of the abstract seems so general as to convey little meaning. I think the last paragraph of conclusions in the text has excellent suggestions for interventions, and these should be the focus of the conclusions in abstract. This will also, in my opinion, lead to more people reading the paper.

Response: Thank you for your positive feedback about our discussion of structural drivers of inequity. Please see our response to Comment 3 by the journal editor. We have kept most of the language used to contextualize our findings and reflect the points we raise in the discussion section about systemic racism and discrimination. At the same time, we are striving to balance this contextual narrative with restating the study findings and keeping the abstract as succinct as possible, as per the journal editor’s comment referenced above.

---

## [Decision Letter · Decision Letter 2]

9 May 2022

PONE-D-21-10832R2The competing effects of racial discrimination and racial identity on the predicted number of days incarcerated in the US: A national profile of Black, Latino/Latina, and American Indian/Alaska Native populationsPLOS ONE

Dear Dr. Pro,

Thank you for submitting your manuscript to PLOS ONE. After careful consideration, we feel that it has merit but does not fully meet PLOS ONE’s publication criteria as it currently stands. Therefore, we invite you to submit a revised version of the manuscript that addresses the points raised during the review process. Please address the following issues:Educational attainment categories: In your last reply to the AE/Reviewers, you have stated that 'some college' category of the educational attainment refers to those who had "some experience with college coursework but did not finish". If so then this category could be better reported as 'college student', if studying at the time of completing the survey or 'college dropouts' if they had left the college without completing/graduating. Please address this issue in the text as well as Tables including supplemental material.Tables 1 and 2: Please shift the column 'Total' as the last column on the right hand side because the focus of the study is on the ethnic/racial groups and not on the total/aggregate of the groups. Tertiles: Three categories of tertiles: low, mid, and high have been used/reported in the text, table 2 and figures 1A-D. Could you please report in the methods section, how low, mid and high tertiles were determined and what values are covered by each of these three tertile categories?. Please submit your revised manuscript by Jun 23 2022 11:59PM. If you will need more time than this to complete your revisions, please reply to this message or contact the journal office at plosone@plos.org. Please include the following items when submitting your revised manuscript:A rebuttal letter that responds to each point raised by the academic editor and reviewer(s). You should upload this letter as a separate file labeled 'Response to Reviewers'.A marked-up copy of your manuscript that highlights changes made to the original version. You should upload this as a separate file labeled 'Revised Manuscript with Track Changes'.An unmarked version of your revised paper without tracked changes. You should upload this as a separate file labeled 'Manuscript'.If applicable, we recommend that you deposit your laboratory protocols in protocols.io to enhance the reproducibility of your results. Protocols.io assigns your protocol its own identifier (DOI) so that it can be cited independently in the future. For instructions see: https://journals.plos.org/plosone/s/submission-guidelines#loc-laboratory-protocols. Additionally, PLOS ONE offers an option for publishing peer-reviewed Lab Protocol articles, which describe protocols hosted on protocols.io. Read more information on sharing protocols at https://plos.org/protocols?utm_medium=editorial-email&utm_source=authorletters&utm_campaign=protocols.

We look forward to receiving your revised manuscript.

Kind regards,

Syed Ghulam Sarwar Shah, M.B.B.S., M.A., M.Sc., Ph.D.

Academic Editor

PLOS ONE

Journal Requirements:

Additional Editor Comments (if provided):

Thanks for submitting your revised manuscript R2. Two external reviewers and the academic editor have raised some minor issues. Please address these issues carefully and submit the revised manuscript.

Reviewers' comments:

Reviewer's Responses to Questions

**Comments to the Author**

1. If the authors have adequately addressed your comments raised in a previous round of review and you feel that this manuscript is now acceptable for publication, you may indicate that here to bypass the “Comments to the Author” section, enter your conflict of interest statement in the “Confidential to Editor” section, and submit your "Accept" recommendation.

Reviewer #2: All comments have been addressed

Reviewer #3: (No Response)

2. Is the manuscript technically sound, and do the data support the conclusions?

Reviewer #2: (No Response)

Reviewer #3: (No Response)

3. Has the statistical analysis been performed appropriately and rigorously? 

Reviewer #2: (No Response)

Reviewer #3: (No Response)

4. Have the authors made all data underlying the findings in their manuscript fully available?

Reviewer #2: (No Response)

Reviewer #3: (No Response)

5. Is the manuscript presented in an intelligible fashion and written in standard English?

Reviewer #2: (No Response)

Reviewer #3: (No Response)

6. Review Comments to the Author

Reviewer #2: I have one minor and optional suggestion for a wording change: When you define the variable about education, the precise definition of "some college" and "college graduate" remains unclear. The issues are that one can graduate from a junior college after 2 years; and that college graduate also have some college. So it might be useful to say what you said in the response note, that college grads means undergraduate degree.

Reviewer #3: This manuscript investigates the association between racial discrimination and racial identity with incarceration risk. Overall data analysis sounds fine. I have minor comments and questions.

Page 7, line 18, “We grouped discrimination scores (0-6) and identity scores (0-8)” should be revised as “We grouped discrimination scores (0-24) and identity scores (0-48)”.

Page 13, Supplemental Tables 1 and 2 are mentioned, but nowhere can find them.

Are there any correlations and/or interactions between Racial discrimination and racial identity?

7. PLOS authors have the option to publish the peer review history of their article (what does this mean?). If published, this will include your full peer review and any attached files.

Reviewer #2: **Yes: **Samuel R Friedman

Reviewer #3: No

---

## [Author Response · Author response to Decision Letter 2]

10 May 2022

Comments from the editor

Comment #1: In your last reply to the AE/Reviewers, you have stated that 'some college' category of the educational attainment refers to those who had "some experience with college coursework but did not finish". If so then this category could be better reported as 'college student', if studying at the time of completing the survey or 'college dropouts' if they had left the college without completing/graduating. Please address this issue in the text as well as Tables including supplemental material.

Response: We appreciate the opportunity to clarify how this variable was created and defined. Please see our response to Comment #4. We have added several sentences to the methods section more clearly outline the specific survey response options that were included in our condensed analytic variable. We have also changed the names of the variable levels to more accurately reflect the highest level of education attained by each survey respondent. We updated Table 1 with the new labels. Table 2 and the supplemental material did not reference the education variable levels. 

Comment #2: Tables 1 and 2: Please shift the column 'Total' as the last column on the right hand side because the focus of the study is on the ethnic/racial groups and not on the total/aggregate of the groups. 

Response: We have shifted the total column to the right hand side for both Table 1 and Table 2. 

Comment #3: Three categories of tertiles: low, mid, and high have been used/reported in the text, table 2 and figures 1A-D. Could you please report in the methods section, how low, mid and high tertiles were determined and what values are covered by each of these three tertile categories? 

Response: We have added a paragraph in the methods section defining how tertiles are created based on the distributions of the identity and discrimination scores around the 33rd and 66th percentiles. We have also included the range of identity and discrimination scores for each of the tertile groups. 

Reviewer 2

Comment #4: I have one minor and optional suggestion for a wording change: When you define the variable about education, the precise definition of "some college" and "college graduate" remains unclear. The issues are that one can graduate from a junior college after 2 years; and that college graduate also have some college. So it might be useful to say what you said in the response note, that college grads means undergraduate degree.

Response: We appreciate the opportunity to clarify how we are defining education, as we agree that the current wording is confusing. While education is not a focus of this study, it acts as an important confounder in our models and we believe it is important for readers to understand how this variable was constructed. We have added text in the methods section defining each of the survey options that we combined to create the new condensed analytic variable. We have also edited the labels for each level to more clearly articulate the highest level of education attained. The new labels are 1) less than high school, 2) high school completion or GED, 3) some college but did not graduate, or received an associate’s or technical degree, and 4) college completion, bachelor’s degree or higher. 

Reviewer 3

Comment #5: Page 7, line 18, “We grouped discrimination scores (0-6) and identity scores (0-8)” should be revised as “We grouped discrimination scores (0-24) and identity scores (0-48)”.

Response: We have corrected this typo. 

Comment #6: Page 13, Supplemental Tables 1 and 2 are mentioned, but nowhere can find them.

Response: Supplemental tables 1 and 2 were included as attachments in the PLOS online application portal. We followed specific instructions not to embed supplemental material in the main document text. I can confirm that they are also included as attachments in this current resubmission. If the supplemental tables do not appear on your end again, please reach out to the academic editor. 

Comment #7: Are there any correlations and/or interactions between racial discrimination and racial identity?

Response: This is a great question and we appreciate your interest in our research. We believe that testing for correlation or an interaction between identity and discrimination is beyond the scope of the current study, but is very much of interest to us for future research endeavors. For this study, we are specifically interested in the main effects of identity and discrimination on incarceration in a fully adjusted model. Testing the hypothesis that the effect of identity on incarceration is conditional on the value of discrimination would be a valuable contribution to the social sciences and criminal justice literature. At the same time and respectfully, we have opted to make no changes to our statistical model and consider your question about alternative methods as positive motivation to continue this line of research.

---

## [Decision Letter · Decision Letter 3]

13 May 2022

The competing effects of racial discrimination and racial identity on the predicted number of days incarcerated in the US: A national profile of Black, Latino/Latina, and American Indian/Alaska Native populations

PONE-D-21-10832R3

Dear Dr. Pro,

We’re pleased to inform you that your manuscript has been judged scientifically suitable for publication and will be formally accepted for publication once it meets all outstanding technical requirements.

Kind regards,

Syed Ghulam Sarwar Shah, M.B.B.S., M.A., M.Sc., Ph.D.

Academic Editor

PLOS ONE

Additional Editor Comments (optional):

Thanks for addressing issues raised by the academic editor and reviewers.

However, I am not aware of the US educational system so I am not sure whether "attended but did not finish graduate school' is correct in the following statement (lines 14-16 on page 7 of article file with track changes). This statement needs to be checked at the article proof checking stage.

"College graduates were defined as those who received a bachelor’s degree, attended but did not finish graduate school, or completed a master’s degree or equivalent or another higher graduate degree."

Reviewers' comments:

Reviewer's Responses to Questions

**Comments to the Author**

1. If the authors have adequately addressed your comments raised in a previous round of review and you feel that this manuscript is now acceptable for publication, you may indicate that here to bypass the “Comments to the Author” section, enter your conflict of interest statement in the “Confidential to Editor” section, and submit your "Accept" recommendation.

Reviewer #2: All comments have been addressed

Reviewer #3: All comments have been addressed

2. Is the manuscript technically sound, and do the data support the conclusions?

Reviewer #2: (No Response)

Reviewer #3: (No Response)

3. Has the statistical analysis been performed appropriately and rigorously? 

Reviewer #2: (No Response)

Reviewer #3: (No Response)

4. Have the authors made all data underlying the findings in their manuscript fully available?

Reviewer #2: (No Response)

Reviewer #3: (No Response)

5. Is the manuscript presented in an intelligible fashion and written in standard English?

Reviewer #2: (No Response)

Reviewer #3: (No Response)

6. Review Comments to the Author

Reviewer #2: (No Response)

Reviewer #3: (No Response)

7. PLOS authors have the option to publish the peer review history of their article (what does this mean?). If published, this will include your full peer review and any attached files.

Reviewer #2: **Yes: **Samuel R Friedman

Reviewer #3: No

---

## [Editor Report · Acceptance letter]

18 May 2022

PONE-D-21-10832R3 

The competing effects of racial discrimination and racial identity on the predicted number of days incarcerated in the US: A national profile of Black, Latino/Latina, and American Indian/Alaska Native populations 

Dear Dr. Pro:

I'm pleased to inform you that your manuscript has been deemed suitable for publication in PLOS ONE. Congratulations! Your manuscript is now with our production department. 

Kind regards, 

on behalf of

Dr. Syed Ghulam Sarwar Shah 

Academic Editor

PLOS ONE